# Design, Synthesis and Evaluation of Novel Trichloromethyl Dichlorophenyl Triazole Derivatives as Potential Safener

**DOI:** 10.3390/biom9090438

**Published:** 2019-09-01

**Authors:** Ke-Liang Guo, Li-Xia Zhao, Zi-Wei Wang, Shu-Zhe Rong, Xiao-Lin Zhou, Shuang Gao, Ying Fu, Fei Ye

**Affiliations:** Department of Applied Chemistry, College of Science, Northeast Agricultural University, Harbin 150030, China

**Keywords:** trichloromethyl dichlorobenzene triazole compounds, synthesis, herbicide detoxification, safener activity

## Abstract

The dominance of safener can unite with herbicides acquiring the efficient protection of crop and qualifying control of weeds in agricultural fields. In order to solve the crop toxicity problem and exploit the novel potential safener for fenoxaprop-*P*-ethyl herbicide, a series of trichloromethyl dichlorobenzene triazole derivatives were designed and synthesized by the principle of active subunit combination. A total of 21 novel substituted trichloromethyl dichlorobenzene triazole compounds were synthesized by substituted aminophenol and amino alcohol derivatives as the starting materials, using cyclization and acylation. All the compounds were unambiguously characterized by IR, ^1^H-NMR, ^13^C-NMR, and HRMS. A greenhouse bioassay indicated that most of the title compounds could protect wheat from injury caused by fenoxaprop-*P*-ethyl at varying degrees, in which compound **5o** exhibited excellent safener activity at a concentration of 10 μmol/L and was superior to the commercialized compound fenchlorazole. A structure–activity relationship for the novel compounds was determined, which demonstrated that those compounds containing benzoxazine groups showed better activity than that of oxazole-substituted compounds. Introducing a benzoxazine fragment and electron-donating group to specific positions could improve or maintain the safener activity for wheat against attack by the herbicide fenoxaprop-*P*-ethyl. A molecular docking model suggested that a potential mechanism between **5o** and fenoxaprop-*P*-ethyl is associated with the detoxication of the herbicide. Results from the present work revealed that compound **5o** exhibited good crop safener activities toward wheat and could be a promising candidate structure for further research on wheat protection.

## 1. Introduction

Herbicides are a major tool for controlling weeds to protect crop yields. From the mid-1980s, more than 140 new herbicide active ingredients were commercialized and played an indispensable role in the rapid development of agriculture and rapid economic growth [1]. However, many were used for years and then eliminated from the market as unfavorable toxicology became apparent or safer and more effective replacements were found [2]. New challenges, such as herbicide toxicity, weed resistance, low crop selectivity, and high cost of discovery, production, and registration of herbicides, have been raised over the years [3]. Therefore, the design of more potent, selective, environmentally friendly and cost-effective herbicides is one of the main the focus on pesticide development [4]. In recent years, herbicides have entered the stage of efficient development and constantly updated varieties [5]. Fenoxaprop-*P*-ethyl (FE) is widely used to selectively control gramineous weeds in wheat fields and as an aryloxyphenoxypropionate postemergence herbicide inhibiting fatty acid synthesis in grasses through inhibition of acetyl-CoA carboxylase (ACCase, EC. 6.4.1.2) [6]. ACCase, the main target of FE, exerts a particularly important role in catalyzing fatty acid biosynthesis. Once initiated, the reaction commits malonyl-CoA to produce fatty acids. In plants, this reaction occurs mainly in plastids and, when inhibited by FE, eventually leads to plant leaf chlorosis death [7]. However, extensive application with the use of herbicides has increased phytotoxicity to the crop plants [8]. The pressure of multiple herbicides can lead to major practical problems, particularly because herbicide toxicity results in dramatic crop yield losses. Thus, it would be urgently necessary to investigate and develop potent safeners for obtaining crop safety and yield.

Herbicide safeners are a group of synthetic compounds that could protect selected plants against herbicide injury without decreasing herbicidal activity to target weeds. Alternatively, herbicide safeners may effectively improve herbicide selectivity [9,10,11]. Nearly 20 safeners have subsequently been developed and commercialized by famous agrochemical companies [12]. Fenchlorazole, one of the triazole herbicide safeners developed by Hoechst AG Company, has received much attention on account of its promising protection of wheat against FE [13]. It may enhance the metabolism of FE, not only by stimulating the activity of the key enzyme P450-dependent monooxygenase but also, by activating isozymes in the detoxification of FE [14]. The presence of the safener also enhances the activity of antiretroviral, glutathione-S-transferase (GST) and promotes the metabolism of herbicides [15]. Benoxacor, a chloroacetamide herbicide safener, can also protect crops against herbicides [16]. It protects maize from chloroacetanilide herbicides (e.g., alachlor) by increasing the expression of GST, which efficiently catalyzes the conjugation of herbicides with glutathione (GSH) [17].

The substructure combination method is very important in the search for bioactive materials, and it has been continuously used in the development of novel pesticides [18,19,20]. Recently, many successful cases have been reported. For example, isoxadifen-ethyl is developed as a herbicide safener with strong protection of rice by combining with the active structure of 5-phenyl-4,5-dihydroisoxazole-3-ethyl ester and the safener diphenyl acid (Scheme 1) [21]. Diclofop-methyl, developed by Farbwerke Hoechst Company, is designed by pharmacophore combination of phenoxycarboxylic acid and diphenyl ether herbicides [22]. As we noted above and continuous of our research on the design of nitrogen-containing heterocyclic safeners [23,24,25,26], a series of the new safener skeleton structure compounds are designed based on active substructure combinations retaining the trichloromethyl dichlorobenzene triazole of fenchlorazole and the nitrogen-containing heterocyclic ring of benoxacor or R-28725 as the parent skeleton structure (Scheme 2). Herein, we report the synthesis of a series of novel substituted trichloromethyl dichlorobenzene triazole compounds via cyclization and acylation without any expensive reagent or catalyst. They are assessed as safeners to protect wheat against the herbicide fenoxaprop-*P*-ethyl. Furthermore, molecular docking analyses are extensively performed on the target compound to identify the possible detoxification mechanism for their safener potency.

## 2. Materials and Methods

### 2.1. Chemicals

The melting points of target compounds were measured using a Beijing Taike melting point apparatus (X-4) (Beijing Tech Instrument Co. Ltd., Beijing, China) and were uncorrected. Infrared (IR) spectra were recorded on a Bruker ALPHA-T spectrometer (BRUKER Inc., Beijing, China) using KBr disks. ^1^H and ^13^C-NMR nuclear magnetic resonance (NMR) spectra were obtained using a Bruker AV-400 spectrometer (BRUKER Inc., Beijing, China) in CDCl_3_ solution with tetramethylsilane as the internal standard. The high-resolution mass spectrometry (HRMS) was recorded on micrOTOF-Q II 10410 spectrometer of Bruker (BRUKER Inc., Beijing, China) and FT-ICR MS spectrometer of Bruker (BRUKER Inc., Beijing, China). The single-crystal diffraction experiment was carried out on a Rigaku R-AXIS RAPID area-detector diffractometer (Rigaku Company, Tokyo, Japan). All solvents and reagents were of analytical reagent grade. Column chromatography purification was carried out on silica gel.

### 2.2. General Procedure for the Synthesis of Compound (***3***)

1-(2,4-Dichlorobenzyl)-5-trichloromethyl-1,2,4-triazole-3-carboxylate **1** (10 mmol) was dissolved in EtOH (20 mL) and NaOH aq. (1 mol/L, 10 mL), and the mixture was stirred at 25 °C for 5 h. Next, distilled water (80 mL) was added into the mixture. The reaction mixture was then acidified by concentrated HCl solution to pH = 1 to induce precipitation. The precipitate was collected through vacuum filtration, washed with water and dried to afford product **2** as white solid without purification. A mixture of compound **2** (37 mmol), sulfoxide chloride (137 mmol, 10 mL) and CH_2_Cl_2_ (20 mL) was stirred for 1 h at 40 °C. Then, five drops of *N*,*N*-dimethylformamide (DMF) was added. After reaction completed, the resulting mixture was concentrated in vacuum to give compound **3** as pale yellow solid (yield 87%), which was used in the next step without further purification.

### 2.3. General Procedure for the Synthesis of Intermediates Compounds (***4***)

The preparation of compounds **4a**–**4u** were conducted as the method reported by references [27,28,29,30,31]. Substituted *o*-aminophenol (27 mmol) was added to a stirred solution of DMSO (40 mL) and K_2_CO_3_ (7.5 g, 54 mmol) of substituted dibromoethane (27 mmol) or (CH_2_Cl_2_, 41 mmol for **4p**–**4q**) at reflux. The mixture was heated under reflux for 6–8 h until the reaction was completed (TLC monitored). The resulting mixture was then filtered, and the filtrate was concentrated under reduced pressure to obtain compounds **4a**–**4q**.

Ketone (26 mmol) and substituted *o*-amino alcohol (26 mmol) were dissolved in toluene (30 mL). After that, the reaction mixture was stirred for another 8 min with microwave irradiation (85 °C, 800 W) to obtain compounds **4r**–**4u**. All the compounds **4a**–**4u** were directly subjected to the next reaction step.

### 2.4. General Procedure for the Synthesis of Title Compounds (***5***)

Compound **3** (15 mmol) and appropriate compound **4** (12.5 mmol) were added to DMAC (30 mL) in succession and the mixture was stirred at room temperature for 2 h. NaOH aq. (2 mol/L, 3 mL) was added to the mixture and the reaction was monitored by TLC. The solvent was evaporated, and the residue was dissolved with ethyl acetate, washed with water, dried and concentrated. The crude product was purified on silica gel with petroleum ether/ethyl acetate (6:1, *v*/*v*) to obtain the target compounds.

*(1-(2,4-Dichlorophenyl)-5-(trichloromethyl)-1H-1,2,4-triazol-3-yl)(3-methyl-2,3-dihydro-4H-benzo[b][1,4]oxazin-4-yl)methanone* (**5a**). White Solid. Yield: 66%. m.p. 204–205 °C. IR (KBr): *ν* (cm^−1^) 3112–2892 (-CH_3_, -CH_2_, =CH), 1662 (C=O), 1587–1500 (C=C); ^1^H-NMR (400 MHz, CDCl_3_) *δ*: 7.60–6.91 (m, 7H, ArH), 4.34–4.21 (m, 3H, CH_2_-CH), 1.40–1.37 (d, *J* = 8.4 Hz, 3H, C-CH_3_); ^13^C-NMR (100 MHz, CDCl_3_) *δ*: 182.90, 155.73, 155.09, 145.89, 145.77, 138.19, 133.99, 133.42, 130.82, 130.59, 127.85, 126.26, 124.46, 122.68, 120.78, 117.09, 85.26, 69.75, 15.65; HRMS (ESI) calcd. for C_19_H_13_Cl_5_N_4_O_2_ [M + H]^+^ 504.9554; found 504.9556 (see Appendix A).

*(6-Chloro-3-methyl-2,3-dihydro-4H-benzo[b][1,4]oxazin-4-yl)(1-(2,4-dichlorophenyl)-5-(trichloromethyl)-1H-1,2,4-triazol-3-yl) methanone* (**5b**). White Solid. Yield: 55%. m.p. 205–207 °C. IR (KBr): ν (cm^−1^) 3092–2896 (-CH_3_, -CH_2_, =CH), 1671 (C=O), 1579–1493 (C=C); ^1^H-NMR (400 MHz, CDCl_3_) δ: 7.59–7.01 (m, 5H, ArH), 6.88–6.85 (m, 1H, ArH), 4.89–4.21 (m, 3H, CH_2_-CH), 1.38–1.36 (m, 3H, C-CH_3_); ^13^C-NMR (100 MHz, CDCl_3_) δ: 158.33, 158.15, 155.64, 144.50, 138.24, 133.99, 133.44, 130.82, 130.61, 130.53, 127.84, 126.25, 124.10, 123.92, 123.58, 118.18, 85.18, 69.76, 15.56; HRMS (ESI) calcd. for C_19_H_12_Cl_6_N_4_O_2_ [M + H]^+^ 538.9154; found 538.9158.

*(1-(2,4-Dichlorophenyl)-5-(trichloromethyl)-1H-1,2,4-triazol-3-yl)(3,6-dimethyl-2,3-dihydro-4H-benzo[b] [1,4]oxazin-4-yl)methanone* (**5c**). White Solid; Yield: 75%. m.p. 194–196 °C. IR(KBr): ν (cm^−1^) 3092–2929 (-CH_3_, -CH_2_, =CH), 1668 (C=O), 1587–1467 (C=C); ^1^H-NMR (400 MHz, CDCl_3_) δ: 7.81–7.42 (m, 4H, ArH), 6.88–6.79 (m, 2H, ArH), 4.82–4.18 (m, 3H, CH_2_CH), 2.24–2.21 (d, J = 12.0 Hz, 3H, C-CH_3_), 1.38–1.26 (m, 3H, C-CH_3_); ^13^C-NMR(100 MHz, CDCl_3_) δ: 158.39, 158.11, 156.16, 155.88, 154.99, 143.64, 138.15, 133.95, 133.45, 130.83, 130.54, 127.84, 127.08, 124.64, 122.23, 116.76, 85.28, 69.69, 20.87, 15.52; HRMS (ESI) calcd. for C_20_H_15_Cl_5_N_4_O_2_ [M + H] ^+^ 518.9638; found 518.9632.

*(6,8-Dichloro-3-methyl-2,3-dihydro-4H-benzo[b][1,4]oxazin-4-yl)(1-(2,4-dichlorophenyl)-5-(trichloromethyl)-1H-1,2,4-triazol-3-yl)methanone* (**5d**). White Solid; Yield: 50%. m.p. 195–196 °C. IR(KBr): ν (cm^–1^) 3092–2966 (-CH_3_, -CH_2_, =CH), 1672 (C=O), 1570–1479 (C=C); ^1^H-NMR (400 MHz, CDCl_3_) δ: 7.78–7.43 (m, 5H, ArH), 4.27–4.17(m, 3H, CH_2_CH), 1.41–1.38(d, J = 10.4Hz, 3H, C-CH_3_); ^13^C-NMR (100 MHz, CDCl_3_) δ: 158.29, 155.71, 155.09, 145.77, 138.17, 133.96, 133.42, 130.83, 130.57, 127.83, 126.26, 124.44, 122.66, 120.76, 117.09, 85.24, 69.73, 59.43, 15.63; HRMS (ESI) calcd. for C_19_H_11_Cl_7_N_4_O_2_ [M + H]^+^ 572.8702; found 572.8707.

*(6-Bromo-3-methyl-2,3-dihydro-4H-benzo[b][1,4]oxazin-4-yl)(1-(2,4-dichlorophenyl)-5-(trichloromethyl)-1H-1,2,4-triazol-3-yl)methanone* (**5e**). White Solid; Yield: 51%. m.p. 214–215 °C. IR (KBr): ν (cm^−1^) 3089–2868 (-CH_3_, -CH_2_, =CH), 1596 (C=O), 1497–1452 (C=C); ^1^H-NMR (400 MHz, CDCl_3_) δ: 7.60–6.80 (m, 6H, ArH), 4.87–4.22 (d, J = 32Hz, 3H, CH_2_CH), 1.28–1.26 (d, J = 8.0Hz, 3H, C-CH_3_); ^13^C-NMR (100 MHz, CDCl_3_) δ: 158.33, 158.15, 155.63, 155.42, 144.50, 138.23, 133.99, 133.43, 130.83, 130.60, 127.82, 126.24, 124.08, 123.57, 118.16, 85.18, 69.74, 51.14, 15.55; HRMS (ESI) calcd. for C_19_H_12_Cl_5_BrN_4_O_2_ [M + H]^+^ 604.8479; found 604.8478.

*(1-(2,4-Dichlorophenyl)-5-(trichloromethyl)-1H-1,2,4-triazol-3-yl)(3,7-dimethyl-2,3-dihydro-4H-benzo[b][1,4]oxazin-4-yl)methanone* (**5f**). White Solid; Yield: 77%. m.p. 203–205 °C. IR (KBr): ν (cm^−1^) 3069–2880 (-CH_3_, -CH_2_, =CH), 1661 (C=O), 1583–1505 (C=C); ^1^H-NMR (400 MHz, CDCl_3_) δ: 7.61–7.44 (m, 6H, ArH), 4.29–4.19 (m, 3H, CH_2_CH), 1.44–1.40 (m, 6H, C-CH_3_); ^13^C-NMR (100 MHz, CDCl_3_) δ: 158.10, 155.80, 155.00, 154.61, 145.56, 138.12, 136.36, 133.94, 133.45, 130.82, 127.77, 124.11, 121.63, 120.05, 117.27, 85.25, 69.67, 53.40, 29.68, 20.92; HRMS (ESI) calcd. for C_20_H_15_Cl_5_N_4_O_2_ [M + H] ^+^ 518.9710; found 518.9712.

*(1-(2,4-Dichlorophenyl)-5-(trichloromethyl)-1H-1,2,4-triazol-3-yl)(6-methoxy-3-methyl-2,3-dihydro-4H-benzo[b][1,4]oxazin-4-yl)methanone* (**5g**). White Solid; Yield: 70%. m.p. 210–212 °C. IR (KBr): ν (cm^−1^) 3052–2940 (-CH_3_, -CH_2_, =CH), 1666 (C=O), 1598–1509 (C=C); ^1^H-NMR (400 MHz, CDCl_3_) δ: 7.60–7.43 (m, 3H, ArH), 6.85–6.67 (m, 2H, ArH), 4.27–4.17 (m, 3H, CH_2_CH), 3.71 (s, 3H, O-CH_3_), 1.41–1.38 (d, J = 12.0Hz, 3H, C-CH_3_); ^13^C-NMR (100 MHz, CDCl_3_) δ: 158.46, 155.78, 154.73, 153.21, 139.88, 138.17, 134.02, 130.79, 130.78, 127.79, 122.79, 122.77, 117.49, 113.33, 108.76, 85.25, 77.23, 69.57, 69.47, 55.84; HRMS(ESI) calcd. for C_20_H_15_Cl_5_N_4_O_3_ [M + H]^+^ 534.9660; found 534.9662.

*(6-(Tert-butyl)-3-methyl-2,3-dihydro-4H-benzo[b][1,4]oxazin-4-yl)(1-(2,4-dichlorophenyl)-5-(trichloromethyl)-1H-1,2,4-triazol-3-yl)methanone* (**5h**). White Solid; Yield: 80%. m.p. 212–214 °C. IR (KBr): ν (cm^−1^) 3089–2874 (-CH_3_, -CH_2_, =CH), 1667 (C=O), 1585–1504 (C=C); ^1^H-NMR (400 MHz, CDCl_3_) δ: 7.62–6.84 (m, 6H, ArH), 4.40–4.05 (m, 3H, CH_2_CH), 1.49–1.22 (m, 12H, 4CH_3_); ^13^C-NMR (100 MHz, CDCl_3_) δ: 157.93, 156.21, 154.72, 143.59, 143.43, 138.09, 134.02, 133.89, 133.44, 130.75, 130.68, 130.58, 130.56, 127.72, 123.35, 121.48, 116.32, 85.27, 69.75, 58.46, 34.33, 31.49, 31.45; HRMS (ESI) calcd. for C_23_H_21_Cl_5_N_4_O_2_ [M + H]^+^ 561.0180; found 561.0173.

*(1-(2,4-Dichlorophenyl)-5-(trichloromethyl)-1H-1,2,4-triazol-3-yl)(6-ethyl-3-methyl-2,3-dihydro-4H-benzo [b][1,4]oxazin-4-yl)methanone* (**5i**). White Solid; Yield: 82%. m.p. 199–201 °C. IR (KBr): ν (cm^−1^) 2973–2870 (-CH_3_, -CH_2_, =CH), 1666 (C=O), 1587–1496 (C=C); ^1^H-NMR (400 MHz, CDCl_3_) δ: 7.60–6.83 (m, 6H, ArH), 4.30–4.19 (m, 3H, CH_2_-CH), 2.53 (s, 2H, C-CH_2_), 1.39–1.17 (m, 6H, 2CH_3_); ^13^C-NMR (100 MHz, CDCl_3_) δ: 158.05, 154.93, 154.60, 143.76, 138.12, 133.94, 133.92, 133.48, 133.42, 130.77, 130.58, 130.54, 127.78, 125.82, 123.45, 122.28, 116.77, 116.72, 85.27, 28.22, 15.77; HRMS (ESI) calcd. for C_21_H_17_Cl_5_N_4_O_2_ [M + H]^+^ 532.9867; found 532.9873.

*(1-(2,4-Dichlorophenyl)-5-(trichloromethyl)-1H-1,2,4-triazol-3-yl)(2,3-dimethyl-2,3-dihydro-4H-benzo[b][1,4]oxazin-4-yl)methanone* (**5j**). White Solid; Yield: 79%. m.p. 199–200 °C. IR (KBr): ν (cm^−1^) 3092–2884 (-CH_3_, -CH, =CH), 1665 (C=O), 1596–1493 (C=C); ^1^H-NMR (400 MHz, CDCl_3_) δ: 7.62–6.90 (m, 7H, ArH), 4.48 (s, 2H, CH-CH), 1.57–1.23 (m, 6H, 2CH_3_); ^13^C-NMR (100 MHz, CDCl_3_) δ: 158.25, 155.74, 146.67, 146.56, 146.51, 140.01, 138.12, 130.81, 130.53, 127.81, 127.72, 126.26, 126.16, 116.84, 85.23, 68.51, 53.85, 53.39, 30.91, 17.76; HRMS (ESI) calcd. for C_20_H_15_Cl_5_N_4_O_2_ [M + H]^+^ 518.9710; found 518.9699.

*(1-(2,4-Dichlorophenyl)-5-(trichloromethyl)-1H-1,2,4-triazol-3-yl)(2,3,6-trimethyl-2,3-dihydro-4H-benzo[b][1,4]oxazin-4-yl)methanone* (**5k**). White Solid; Yield: 77%. m.p. 198–200 °C. IR (KBr): ν (cm^−1^) 3012–2854 (-CH_3_, -CH, =CH), 1662 (C=O), 1502–1496 (C=C); ^1^H-NMR (400 MHz, CDCl_3_) δ: 7.62–6.82 (m, 6H, ArH), 4.53–4.29 (m, 2H, CH-CH), 2.28–2.07 (m, 3H, ArCH_3_), 1.38–1.15 (m, 6H, C-CH_3_); ^13^C-NMR (100 MHz, CDCl_3_) δ: 158.25, 155.05, 151.29, 146.51, 138.12, 133.92, 133.43,130.81, 130.53, 127.81, 126.26, 126.16, 124.42, 122.22, 116.84, 92.36, 85.23, 68.07, 30.91, 20.84, 17.76; HRMS (ESI) calcd. for C_21_H_17_Cl_5_N_4_O_2_ [M + H]^+^ 532.9867; found 532.9851.

*(6,8-Dichloro-2,3-dimethyl-2,3-dihydro-4H-benzo[b][1,4]oxazin-4-yl)(1-(2,4-dichlorophenyl)-5-(trichloromethyl)-1H-1,2,4-triazol-3-yl)methanone* (**5l**). White Solid; Yield: 55%. m.p. 194–196 °C. IR (KBr): ν (cm^−1^) 3050–2874 (-CH_3_, -CH, =CH), 1661 (C=O), 1586–1469 (C=C); ^1^H-NMR (400 MHz, CDCl_3_) δ: 7.59–6.80 (m, 5H, ArH), 4.40–4.10 (m, 2H, CH-CH), 2.33–2.06 (m, 3H, C-CH_3_), 1.59–0.87 (m, 3H, C-CH_3_); ^13^C-NMR (100 MHz, CDCl_3_) δ: 171.15, 158.59, 155.73, 154.79, 144.45, 138.16, 133.92, 133.38, 130.81, 130.55, 127.80, 127.27, 124.57, 123.86, 117.01, 85.21, 66.05, 60.40, 20.76, 14.22; HRMS (ESI) calcd. for C_20_H_13_Cl_7_N_4_O_2_ [M + H]^+^ 587.8829; found 587.8819.

*(1-(2,4-Dichlorophenyl)-5-(trichloromethyl)-1H-1,2,4-triazol-3-yl)(6-ethyl-2,3-dimethyl-2,3-dihydro-4H-benzo[b][1,4]oxazin-4-yl)methanone* (**5m**). White Solid; Yield: 81%. m.p. 213–215 °C. IR (KBr): ν (cm^−1^) 3089–2870 (-CH_3_, -CH_2_, =CH), 1655 (C=O), 1586–1497 (C=C); ^1^H-NMR (400 MHz, CDCl_3_) δ: 8.00–6.81 (m, 6H, ArH), 4.93–4.28 (m, 2H, CH-CH), 2.60–2.57 (m, 2H, C-CH_2_), 1.42–1.20 (m, 9H, 3CH_3_); ^13^C-NMR (100 MHz, CDCl_3_) δ: 158.10, 156.14, 155.80, 155.00, 154.61, 145.56, 145.49, 138.12, 136.36, 133.94, 133.45, 130.82, 130.52, 127.77, 124.11, 120.05, 117.27, 85.25, 29.68, 20.92, 15.42, 13.68; HRMS (ESI) calcd. for C_22_H_19_Cl_5_N_4_O_2_ [M + H]^+^ 547.0023; found 547.0009.

*(1-(2,4-Dichlorophenyl)-5-(trichloromethyl)-1H-1,2,4-triazol-3-yl)(2,3-dihydro-4H-benzo[b][1,4]oxazin-4-yl)methanone* (**5n**). White Solid; Yield: 55%. m.p. 189–191 °C. IR (KBr): *ν* (cm^−1^) 3095–2887 (-CH_3_, -CH_2_, =CH), 1667 (C=O), 1587–1495 (C=C); ^1^H-NMR (400 MHz, CDCl_3_) *δ*: 8.18–6.92 (m, 7H, ArH), 4.43–4.18 (m, 4H, CH_2_−CH_2_); ^13^C-NMR (100 MHz, CDCl_3_) *δ*: 158.35, 155.57, 154.87, 146.62, 138.17, 133.92, 133.33, 130.79, 130.54, 127.79, 126.52, 124.99 123.72, 120.28, 117.35, 85.17, 66.10, 66.06; HRMS (ESI) calcd. for C_18_H_11_Cl_5_N_4_O_2_ [M + H]^+^ 490.9397; found 490.9389.

*(1-(2,4-Dichlorophenyl)-5-(trichloromethyl)-1H-1,2,4-triazol-3-yl)(6-methyl-2,3-dihydro-4H-benzo[b][1,4]oxazin-4-yl)methanone* (**5o**). White Solid; Yield: 78%. m.p. 196–198 °C. IR (KBr): *ν* (cm^−1^) 3087–2928 (-CH_3_, -CH_2_, =CH), 1659 (C=O), 1507–1470 (C=C); ^1^H-NMR (400 MHz, CDCl_3_) *δ*: 7.64–6.81 (m, 6H, ArH), 4.40–4.15 (m, 4H, CH_2_-CH_2_), 2.19 (s, 3H, C-CH_3_); ^13^C-NMR (100 MHz, CDCl_3_) *δ*: 158.35, 155.57, 154.87, 146.62, 138.17, 133.92, 133.33, 130,79, 130.54, 130.54, 127.79, 126.52, 124.99, 123.72, 120.33, 117.35, 85.17, 66.10, 50.45; HRMS (ESI) calcd. for C_19_H_13_Cl_5_N_4_O_2_ [M + H]^+^ 529.5971; found 529.5979.

*Benzo[d]oxazol-3(2H)-yl(1-(2,4-dichlorophenyl)-5-(trichloromethyl)-1H- 1,2,4-triazol-3- yl)methanone* (**5p**). White Solid; Yield: 51%. m.p. 249–250 °C. IR (KBr): *ν* (cm^−1^) 3090–2987 (-CH_2_, =CH), 1704 (C=O), 1541–1458 (C=C); ^1^H-NMR (400 MHz, CDCl_3_) *δ*: 8.56–8.53 (m, 2H, ArH), 7.63–7.28 (m, 5H, ArH), 6.00 (s, 2H, CH_2_); ^13^C-NMR (100 MHz, CDCl_3_) *δ*: 155.06, 154.98, 154.95, 145.53, 138.22, 134.14, 133.43, 130.80, 130.58, 127.78, 127.69, 124.83, 123.50, 120.92, 113.76, 91.73, 85.08; HRMS (ESI) calcd. for C_17_H_9_Cl_5_N_4_O_2_ [M + H]^+^ 476.9241; found 476.9224.

*(1-(2,4-Dichlorophenyl)-5-(trichloromethyl)-1H-1,2,4-triazol-3-yl)(5-methylbenzo[d]oxazol-3(2H)-yl)methanone* (**5q**). White Solid; Yield: 59%. m.p. 202–204 °C. IR (KBr): *ν* (cm^−1^) 3082–2911 (-CH_3_, -CH_2_, =CH), 1620 (C=O), 1519–1441 (C=C); ^1^H-NMR (400 MHz, CDCl_3_) *δ*: 7.62–7.17 (m, 6H, ArH), 4.51 (s, 2H, O-CH_2_), 2.19–1.47 (m, 3H, ArCH_3_); ^13^C-NMR (100 MHz, CDCl_3_) *δ*: 158.24, 155.34, 155.08, 154.94, 141.66, 138.26, 133.89, 133.29, 133.19, 130.73, 130.61, 130.52, 127.87, 126.39, 122.71, 122.25, 74.83, 17.56; HRMS (ESI) calcd. for C_18_H_11_Cl_5_N_4_O_2_ [M + H]^+^ 490.9397; found 490.9389.

*(1-(2,4-Dichlorophenyl)-5-(trichloromethyl)-1H-1,2,4-triazol-3-yl)(1-oxa-4-azaspiro[4.5]decan-4-yl)methanone* (**5r**). White Solid; Yield: 70%. m.p. 185–187 °C. IR (KBr): *ν* (cm^−1^) 3094–2862 (-CH_2_, =CH), 1653 (C=O), 1582–1476 (C=C); ^1^H-NMR (400 MHz, CDCl_3_) *δ*: 7.62–7.28 (m, 3H, ArH), 4.04 (s, 4H, CH_2_-CH_2_), 2.74–1.27 (m, 10H, 5CH_2_; ^13^C-NMR (100 MHz, CDCl_3_) *δ*: 158.46, 156.59, 154.79, 138.17, 134.11, 130.70, 127.70, 127.65, 97.57, 85.10, 61.99, 48.05, 41.97, 31.73, 27.02, 25.02, 24.56, 23.18; HRMS (ESI) calcd. for C_18_H_17_Cl_5_N_4_O_2_ [M + H]^+^ 496.9794; found 496.9799.

*(1-(2,4-Dichlorophenyl)-5-(trichloromethyl)-1H-1,2,4-triazol-3-yl)(2-methyl-1-oxa-4-azaspiro[4,5]decan-4-yl) methanone* (**5s**). White Solid; Yield: 76%. m.p. 196–198 °C. IR (KBr): *ν* (cm^−1^) 3089–2867 (-CH_3_, -CH_2_, =CH), 1650 (C=O), 1503–1478 (C=C); ^1^H-NMR (400 MHz, CDCl_3_) *δ*: 7.61–7.28 (m, 3H, ArH), 4.18–3.46 (m, 3H, CH_2_-CH), 2.80–2.56 (m, 10H, 5CH_2_, 1.36(s, 3H, C-CH_3_); ^13^C-NMR(100 MHz, CDCl_3_) *δ*: 156.54, 155.74, 138.03, 134.07, 133.54, 130.81, 130.55, 127.74, 127.70, 97.89, 85.35, 70.41, 54.68, 33.69, 30.55, 24.61, 23.21, 23.11, 18.00; HRMS (ESI) calcd. for C_19_H_19_Cl_5_N_4_O_2_ [M + H]^+^ 512.9981; found 512.9988.

*(1-(2,4-Dichlorophenyl)-5-(trichloromethyl)-1H-1,2,4-triazol-3-yl)(2,2-dimethyloxazolidin-3-yl)methanone* (**5t**). White Solid; Yield: 54%. m.p. 175–177 °C. IR (KBr): *ν* (cm^−1^) 3069–2888 (-CH_3_, -CH_2_, =CH), 1649 (C=O), 1584–1489 (C=C); ^1^H-NMR (400 MHz, CDCl_3_) *δ*: 7.62–7.42 (m, 3H, ArH), 4.08–3.50 (m, 4H, CH_2_-CH_2_), 2.18 (s, 6H, C-CH_3_); ^13^C-NMR (100 MHz, CDCl_3_) *δ*: 158.46, 156.28, 155.68, 138.08, 134.04, 130.76, 130.55, 127.74, 96.02, 63.82, 61.98, 47.93, 42.39, 30.97, 24.24; HRMS (ESI) calcd. for C_15_H_13_Cl_5_N_4_O_2_ [M + H]^+^ 456.9554; found 456.9548.

*(1-(2,4-Dichlorophenyl)-5-(trichloromethyl)-1H-1,2,4-triazol-3-yl)(2,2,5-trimethyloxazolidin-3-yl)methanone* (**5u**). White Solid; Yield: 67%. m.p. 182–183 °C. IR (KBr): *ν* (cm^−1^) 3089–2888 (-CH_3_, -CH_2_, =CH), 1654 (C=O), 1585–1495 (C=C); ^1^H-NMR (400 MHz, CDCl_3_) *δ*: 7.62–7.43 (m, 3H, Ar-H), 4.34–3.47 (m, 3H, CH_2_-CH), 1.79–1.72 (m, 6H, C-CH_3_), 1.39–1.27 (m, 3H, C-CH_3_); ^13^C-NMR (100 MHz, CDCl_3_) *δ*: 158.37, 154.76, 138.06, 134.09, 133.49, 130.70, 130.53, 127.64, 96.23, 70,77, 67.28, 53.38, 46.93, 30.92, 25.67, 21.09; HRMS (ESI) calcd. for C_16_H_15_Cl_5_N_4_O_2_ [M + H]^+^ 470.9710; found 470.9701.

### 2.5. X-ray Diffraction

A suitable single crystal of compound **5j** (CCDC 1909489) was obtained by dissolving the compound in a solution of ethyl acetate, and the solvent was placed at room temperature to slowly evaporate the solvent in air for several days. Bright white crystals of **5j** (0.13 mm × 0.12 mm × 0.10 mm) were selected and mounted on a Rigaku R-AXIS RAPID area detector diffractometer equipped with graphite-monochromated Mo-*K*α radiation (λ = 0.71073Å); θmax = 27.58. The crystal was kept at 293 K during data collection. Direct methods were used for structure solving by the SHELXS-97 program. A total of 28,327 integrated reflections were collected, and of those, 5128 were unique in the range with R_int_ = 0.0392 and Rsigma = 0.0457. Full-matrix least-squares refinement based on F^2^ employing the weight of ω = 1/[ δ^2^(F_0_^2^) + (0.0470P)^2^ + 2.0812P] gave final values of R_1_ = 0.0334, ωR_2_ = 0.0981 and GOF(F) = 1.066 for 309 variables, 309 parameters and 2363 contributing reflections. Maximum shift/error = 0.003, maximum/minimum residual electron density = 0.396/−0.428 eÅ^−3^.

### 2.6. Biological Assay

The concentration of the safener and compounds applied in the bioassay was determined after a preliminary screening. Wheat seeds (College of Agriculture, Northeast Agricultural University, Harbin, China) were moistened with warm water for 30 min, and then soaked with 0.6% carbendazim for about 30 min. Afterwards, the seeds were washed with distilled water and soaked by title compounds (10 μmol/L) for 12 h, and then germinated for 24 h at 26.5 ± 1 °C. The commercial safener fenchlorazole was chosen as a positive control. The seeds were planted 1.0 cm deep in paper cups, and then sprayed by FE until the two-leaf stage was reached. After 7 days, the effects of the title compounds on the detoxification of FE in soil were evaluated by testing the growth index [32,33].

### 2.7. Computational Methods

The steric structure of compound **5o** and commercial safener fenchlorazole were generated with the sketch module of SYBYL-X 2.0 (Tripos Inc., Saint Louis, MO, USA). Subsequently, the Gasteiger–Huckel charges were calculated, and the tested molecules were optimized. The crystallographic structure of ACCase was obtained from the Protein Data Bank (PDB: 1UYR). Subsequently, docking calculations were performed using the CDOCKER method in Accelrys Discovery Studio 3.0 (BIOVIA Inc., San Diego, CA, USA) [34,35]. Before the docking, the CHARMM force field was added to the protein structure, and other co-crystallized small molecules and water were isolated. After that, the active site was defined, with a subset region of 13.0 Å by the center of the known ligand. In the docking process, the top 10 conformations were reserved for every ligand based on -CDOCKER_ENERGY and the remaining parameters were set to the default values.

## 3. Results and Discussion

### 3.1. Synthesis

The preparation of compounds **3** involved the hydrolysis reaction and acylating chlorination by fenchlorazole-ethyl (Scheme 3). The intermediates **4** were synthesized with substituted aminophenol or amino alcohol derivatives as the starting materials (Scheme 4). The synthetic procedures for the title compounds are depicted in Scheme 5. Compounds **5** were obtained by the compound **3** with compounds **4** (yields in range of 50–82%) (Table 1).

All the title products were characterized by IR, ^1^H-NMR, ^13^C-NMR, and HRMS spectroscopies. Taking compound **5j** as an example. IR analysis showed the existence of a carbonyl group at approximately 1665 cm^−1^ and 1253 cm^−1^ for compound **5j**. In the ^1^H-NMR spectrum analysis, the signals in the region of δ 7.62–6.90 ppm related to benzene ring-bound protons. The signals at δ 1.57–1.23 ppm were characteristic of two methyl groups of the oxazine ring. In the ^13^C-NMR spectra, the signals observed at δ 158.25 ppm were attributed to the carbon of the carbonyl group. The treated HRMS information also corroborated the accuracy of the proposed structures. Meanwhile, the molecular structure of **5j** was additionally determined by X-ray analysis, as seen from Figure 1.

### 3.2. The Structure–Activity Relationships

All compounds **5** (10 μmol/L) were evaluated for their protection of wheat in vivo against FE (Table 2). The growth index was measured after 7 days of treatment including recovery of root length, recovery of plant weight and chlorophyll content.

The herbicide FE was absorbed by the plant and converted into the corresponding acid, which reached the growth point of the plant meristem and the root growth point, affecting plant growth [36,37]. FE provoked a significant chlorosis and inhibition of growth for maize, but recovered when the safener fenchlorazole was included [38]. The chlorophyll content of plant leaves directly affects photosynthesis associated with plant growth [39]. A positive correlation between growth level and recovery of root length, recovery of plant weight and chlorophyll content has been observed in previous research [40]. As seen from Table 2, it was observed that compounds **5** showed different recoveries of wheat growth. Compound **5o** possessed good protection against FE, as indicated by the 70.76% recovery of root length, 143.51% recovery of plant weight, and 99.77% recovery of chlorophyll, respectively, which was superior to the commercial safener fenchlorazole. Compounds **5a**–**5o** showed a more significant improvement of growth index than compounds **5p**–**5u** due to the introduction of substituted benzoxazine. For the preliminary assessment of the substituents on the benzoxazine derivatives, the electron-donating group in the benzoxazine ring was crucial for good safener activity. It was noticeable that compounds **5c** and **5i** exhibited superior detoxification activity compared to compound **5e** due to the replacement of the Br atom with the methyl or ethyl of the substitution at 4-position of benzene ring. Compound **5c** exerted a strong protective effect against FE with recovery rates of root length 71.2% and recovery of plant weight 109.87%, respectively. The result indicated that compounds with a benzoxazine fragment and electron-donating groups were greatly associated with the protective effect of wheat.

An improvement safener activity of **5j**–**5m** against FE was observed relative to the 4-position of the benzene. Additionally, compounds in which substitution at 4-position of benzene ring is by two methyl groups were always favorable to the safener potency than that of one methyl. Most interesting, comparing the protective effects of compounds **5j**–**5m** showed that 4-position substituted with at least one methyl group led to a remarkable improvement in safener activity.

With the above encouraging results, it can be concluded that the activity of compounds with the same parent skeleton was even more related to the various substituents (Figure 2). Moreover, it was found that substitution at the 4-position of benzene was essential for maintaining safener activity. Particularly, the protection of compound **5o** was superior to that of fenchlorazole, which is a commercial safener and effectively reduced injury from FE herbicides.

### 3.3. Molecular Docking Studies

It is reasonable to expect that the physical properties of a molecule on its ability to express activity are important in the selection of potential molecular candidates. Then, the fenoxaprop-*P*-ethyl, fenchlorazole and compound **5o** were compared and analyzed. It was observed that the log *p* and HBAs of compound **5o** were consistent with that of the safener fenchlorazole (Table 3). There was little difference in ARs and RBs between fenchlorazole and compound **5o**. Molecular docking experiment showed that the fenchlorazole and compound **5o** exited multiple interacts with active site such as hydrogen bond and Van der Waals forces etc. However, fenchlorazole and compound **5o** were mainly dominated by hydrogen bonds, therefore the slight difference between ARs and RBs has negligible effect to the safety activity.

To further explore the possible and preliminary effect of the relation mode of compound **5o** involved in FE herbicides, a molecular docking experiment was conducted (Figure 3). In the docking modeling research, the whole molecular skeleton of fenoxaprop acid, in which the ethyl ester is a procide of fenoxaprop acid, could be effectively embedded in the target active site by stronger H-bond interactions with ALA-1627 and ILE-1735, and two π-π interactions with TYR-1738, which gave rise to herbicide activity [41]. Interestingly, both compound **5o** and fenchlorazole could also enter the active channel. In contrast, both compound **5o** and fenchlorazole were located below the active binding site, compared with that of fenoxaprop acid, which just partially obstructs the entrance and interacts with very weak forces such as Van der Waals forces etc. The small substrate had a greater chance to propel itself into the channel leading to the active site and catalyze the active site, preventing the binding of fenoxaprop acid with ACCase [42].

This result showed that the effect of competing with the herbicide at the active site might be ruled out and demonstrated the validity of the previously active fragment we designed, which was closely related to the results of bioassay test. When compound **5o** as safener was used as safener synchronously or before FE, antagonism potentially existed with FE in wheat, which protected the wheat from injury by FE herbicide. This result was similar to that of previous research [43,44], and might be the detoxification mechanism of the novel target compounds.

## 4. Conclusions

In summary, a series of new substituted trichloromethyl dichlorobenzene triazole compounds were rationally designed by the substructure combination method. All the title compounds were confirmed by ^1^H-NMR, ^13^C-NMR, and HRMS. The bioassay showed that most of them exhibited varying amounts of safening activity to wheat from FE. Notably, compound **5o** exhibited better recovery rate of root length, plant height and chlorophyll with 70.76%, 143.51%, and 99.77%, respectively, which was even superior to the commercial safener fenchlorazole. The molecular docking experiment illustrated that the protective effect of compound **5o** might be attributed to antagonism with FE at the active site. The present work indicates that compound **5o** could be used as a candidate for further optimization of a potential safener.

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
