# Peer review of "Design, Synthesis and Evaluation of Novel Trichloromethyl Dichlorophenyl Triazole Derivatives as Potential Safener"

_biomolecules, 2019, doi:10.3390/biom9090438_

Round 1

Reviewer 1 Report

With the current manuscript (biomolecules-570299), a series of synthetic chlorinated triazole derivatives are assayed as potential herbicide safeners. The study was properly conceptualized, the experiments have been well conducted, the results being clearly presented. The relevance and scientific novelty of the current study derives from the protective effects observed with a few synthetic derivatives, particularly 5o, upon wheat exposed to fenchlorazole.

As such, and strictly considering the validity of the experiments and obtained results, I would consider that the current study merits publication after minor revision. That said, there are some issues that should be considered by the authors as below indicated.

Although generally well written, English language editing is mandatory, as evidenced in:

Lines 26-27: Consider “Molecular docking model suggested displayed that a potential mechanism between 5o and fenoxaprop-P-ethyl is associated with the detoxication of herbicide…”.

Lines 39-40: Consider “Therefore, the design of more potent, selective, environmentally friendly and cost-effective herbicides is one of the main focus on pesticide discovery research field development.

Lines 98-100: The sentence is incoherent.

Lines 310-311: Consider “The growth index included recovery of root length, recovery of plant weight and chlorophyll content, were measured after treatment for 7 days.”

Lines 313-314: “…and the root growth point, affecting and the growth status of growth is affected.

Additional examples can be found throughout the whole manuscript, the authors being advised to have the manuscript edited by a native speaker or English editing service. In fact, the conclusion is extremely messy and unclear.

The following corrections should be also considered:

Scheme 2: Revise the structure name to “Fenchlorazole-ethyl”   

Line 283: Revise “oxaole” to “oxazole”.

Line 302: Based on the structure of 5j and associated spectral data, the signals at 1.57-1.23 should correspond to the two methyl substituents and not methylene groups.

Table 1: The substituents (R5) of compound 5n and 5o should be indicated (H) as in the previous examples.

In conclusion, the spectral data seems to corroborate the proposed structures, and the experiments concerning the protective effects are valid, solely a minor revision being required. Nevertheless, I appeal again to the editor to evaluate the potential suitability of the present work and the alignment with the scope of the journal, as the compounds under study correspond to purely synthetic derivatives.  

Reviewer 2 Report

Manuscript biomolecules-570299 entitled „Design, Synthesis and Evaluation of Novel 2 Trichloromethyl Dichlorobenzene Triazole Derivatives as Potential Safener”

The Authors designed and synthesized a series of twenty-one novel substituted trichloromethyl dichlorobenzene triazole compounds and performed evaluation of their bioactivity herbicide safeners.  The most active compound was selected for molecular docking study.

The manuscript requires major changes and corrections. The Authors may consider the following comments:

Considering the fact that the biological studies presented in the paper are only a preliminary assessment of the potential of the compounds obtained as safeners, the Authors should improve the part on the synthesis of new derivatives in the question.

1) Descriptions of the general procedures for the synthesis of compounds 3 and 5 should be carefully corrected.

2) There is a lack of information on the synthesis of compounds designated as 4, which next to compound 3 are reactants in the main stage of the synthesis of the title derivatives 5. The entire commentary on this part is included in the following sentences:

Line 105: The preparation of compounds 4a-4m, 4n-4q and 4r-4u were conducted as the method reported by references [26-28].

Line 271: The intermediates 4 were synthesized with substituted nitroaryl oxygen, aminophenol and amino alcohol derivatives (Scheme 4) - An incomprehensible sentence.

Scheme 4 should be improved and supplemented. References 26-28 do not describe the synthesis of all compounds in this group. It is not known whether the Authors isolated compounds 4 or whether they immediately used them for further reaction. If the compounds were isolated, their full characteristics should be included, i.e. specific references (if they are known and previously described) or a complete set of information (as for compounds of the 5 series).

The summary of physicochemical parameters and spectroscopic data of Series 5 compounds starting from line 116 contains a lot of errors. There is no description of the IR spectra that are included in the supplement.

3) Some of the spectra in the supplement show the impurities present in the samples.

4) The discussion concerning the obtained results is rather poor. In my opinion, the Authors should take into account deeper considerations regarding the current state of knowledge in the field of similar molecules synthesis and structure-activity relationships.

For example, in the Synthesis section, the Authors described the influence of substituents on reactivity of substrates based on the yield of products obtained in the N-acylation reaction. However, they should also cite literature in the subject.

Line 282-285: the description of the data contained in the table should contain the numbers of the compounds concerned,

Line 288: It is inappropriate to use the term "at the R1 position of the benzene" - each structure should be numbered and the numbering of the atoms with which the substituent is connected should be used in the description.

Line 300: IR analysis showed the existence of a carbonyl group at 1665 cm-1 for compound 5j – this is an absolutely incorrect description and, moreover, is the C = O bond vibration band observed only for this compound in the IR spectrum?

The 1H NMR spectrum analysis, the signals in the region of . 7.62-6.90 ppm linked to benzene ring – in 1H NMR spectra, we observe signals from ring-bound protons.

Line 304 and 307: "crystal structure" should be "molecular structure", and in the caption of Fig. 1 information about probability level (%) should be added

5) The section 3.2. The Structure-Activity Relationships should be supplemented with literature information and rewritten due to the large number of errors.

6) The section 3.3. Molecular Docking Studies: “It was observed that the log p, 351 ARs, SA, RBs, HBAs and electronegativity of compound 5o were all consistent with that of the safener fenchlorazole (Table3) - only two values match for both compounds: logp and HBAs.

7) The conclusions need to be rewritten because in their current form they are rather a summary of what was done in the work.

8) Manuscript contains many mistakes and some typographical errors. Each table has different fonts and spacing. The supplement should contain the title of the work and table of contents, which would facilitate the reader to search for spectra of individual compounds..

English and grammar need checking.

Round 2

Reviewer 2 Report

Dear Authors,

The manuscript has been corrected according to most of the comments. However, some corrections are still needed.

In the title, the compound name “Dichlorobenzene” is wrong, and therefore, it should sound: “Design, Synthesis and Evaluation of Novel Trichloromethyl Dichlorophenyl Triazole Derivatives as Potential Safener”.

In the revised version of the manuscript, the letters “H” disappeared on lines 123-261. Although in such a situation it is difficult to determine the correctness of the description of the spectra, some errors can be noted.

The Authors should carefully check all spectral descriptions included in the paper.

The presence of residual signals from solvents used in synthesis or purification is indeed very common and may result, for example, from the formation of solvates. However, after analyzing the spectra of compounds 5a-5f, it can be seen that in many cases, e.g. 5c, 5d and especially 5f, are signals that were incorrectly described or not described at all.
It is not known how were calculated the coupling constants for doublets, which cannot even be called doublets, e.g. 2.21-2.04–1.30(d, J=6.5z, 3, 147 C-C3); (line 147).

Why, such similar structures (5a-5f), which differ only in substituents in the benzene ring, have so different 1H NMR spectra, and the signal from the methyl group in the C3 position of the benzoxazine system is either at δ = 1.37 ppm (5a), or at δ =4.65–4.25ppm (m, 6, 2C3) (5c).

Moreover, it seems that the authors measured the integration of individual signals in a way that matched their interpretation, e.g. „The 1H NMR spectrogram of compound of compound 5f” (suppl., p. 13) – multiplet „4.66–4.06(m, 1, C)” (line 161).

If the proton signals of the described compound overlap with the solvents signasl, the solvents should be changed or the compound further purified. Otherwise, the yields of the reaction cannot be given to the first decimal place, and the yields should not be the basis for discussion on the reactivity of the substrates used. In addition, it is difficult to calculate the yields of reactions whose substrate has not been purified. Therefore, in my opinion, the description of lines 308-323 should be shortened or deleted.

Line 91 – “High-resolution mass spectra were obtained using a Bruker spectrometer.” - This is not enough information on the methodology of measuring mass spectra.

Line 105: reaction yield, please put in brackets

After reaction completed, the resulting mixture was concentrated in vacuum to give compound 3 as  pale yellow solid (yield of  87.4%), which was used in the next step without further purification

Line 107 - I suggest dividing the procedure for obtaining compounds 4 into two parts:

The preparation of compounds 4a-4u were conducted as the method reported by references [27-31].

Compounds 4a-4q

Substituted o-aminophenol  (27 mmol) was added to a stirred solution of DMSO (40 mL), K2CO3 (7.5 g, 54 mmol) in a mixture of and substituted dibromoethane (27 mmol, for 4a-4o) or CH2Cl2 (41 mmol for 4p-4q) at reflux temperature. The  mixture was heated under reflux for 6–8 h until the reaction was completed (TLC monitored). The  resulting mixture was then filtered, and the filtrate was concentrated under reduced pressure to obtain compounds 4a–4q.

Compounds 4r-4u

Ketone (26 mmol) and substituted o-amino alcohol (26 mmol) were added dissolved ? in toluene (30 mL). After that, the reaction mixture was stirred for another 8 min with microwave irradiation (85 °C, 800 W) to obtain compounds 4r–4u.

All compounds 4 possessed ideal yields in range …. and were directly subjected to the next reaction step.

What about toluene, on what basis were the “ideal yields” calculated for compounds 4r-4u?

Line 117 –

Compound 3 (15 mmol) and appropriate compound 4 (12.5 mmol) were added to DMAC (30 mL) in  succession and the mixture was stirred at room temperature for 2 h.

Line 299 –

Compounds 5 were obtained by the acylation of trichloromethyl dichlorobenzene triazole formyl chloride 3 with compounds 4 in DMAC with stirring for 2 h at  (yields of  in range 50.6%–82.6%)

trichloromethyl dichlorobenzene triazole formyl chloride is the wrong name for compound 3. If the authors do not want to use the full chemical name, it is better to write “compound 3” orformyl chloride 3

Line 334 –

Figure 1. Molecular structure of compound 5j and probability level is 30%.

Should be:  Figure 1. Molecular structure of compound 5j. The thermal ellipsoids are shown  at 30% probability level.
